# Landmark Detection Uncertainty as a Reliability Weight for Robust Landmark-based 2D/3D Pelvic Pose Estimation

**Yehyun Suh**[1,2,3] (ORCID)                                   YEHYUN.SUH@VANDERBILT.EDU
[1] *Department of Computer Science, Vanderbilt University, Nashville, TN, USA*
[2] *Vanderbilt Institute of Surgery and Engineering, Nashville, TN, USA*
[3] *Vanderbilt Lab for Immersive AI Translation, Nashville, TN, USA*

**Brayden Schott**[1,2,3]                                   BRAYDEN.J.SCHOTT@VANDERBILT.EDU
**Chou Mo**[4]                                                    ATHENAMO@G.UCLA.EDU
[4] *Department of Mathematics, University of California-Los Angeles, Los Angeles, CA, USA*

**J. Ryan Martin**[5]                                             JOHN.MARTIN@VUMC.ORG
[5] *Department of Orthopaedic Surgery, Vanderbilt University Medical Center, Nashville, TN, USA*

**Daniel Moyer**[*1,2,3]                                       DANIEL.MOYER@VANDERBILT.EDU

**Editors:** Accepted for publication at MIDL 2026

## Abstract

Landmark-based 2D/3D pelvis registration is vulnerable to noisy or ambiguous landmark detections in fluoroscopy, which can destabilize downstream pose estimation. We present an uncertainty-aware registration framework that models epistemic uncertainty in predicted landmarks and incorporates it directly into the Perspective-n-Point formulation. Using Monte Carlo dropout within a U-Net detector, we compute sample-specific per-landmark reliability estimates using the variance of multiple stochastic forward passes. These reliability estimates guide two complementary strategies: continuous weighting, which integrates uncertainty into a weighted PnP optimization, and discrete selection, which removes the most uncertain landmarks during inference. We evaluate the framework on both CT-derived synthetic fluoroscopy and real fluoroscopy from DeepFluoro. Our experiments show that uncertainty provides a principled mechanism for identifying unreliable landmarks and stabilizing pose estimation, enabling more robust registration and establishing a foundation for uncertainty-guided image-guided surgical workflows. Code: https://github.com/yehyunsuh/Uncertainty-Aware-Pelvic-Pose-Estimation

**Keywords:** Uncertainty-Weighted Pose Estimation, Landmark-based 2D/3D Registration, Monte Carlo Dropout, Epistemic Uncertainty Modeling

## 1. Introduction

2D/3D registration is an optimization task that aligns a 2D image with a 3D volume in a common spatial reference frame (Grupp et al., 2018; Unberath et al., 2021). In typical workflows, an X-ray or fluoroscopy image is registered to a CT scan so that 3D anatomical information compensates for the loss of depth and perspective in 2D projections. This paradigm is widely used in image-guided orthopaedic, spine, trauma, and vascular procedures that require precise localization of anatomy, tools, and implants, where fast but

---

[*] Corresponding Author

| Method | Mean Runtime (s) | Median Rot. Error (deg) | Median Trans. Error (mm) |
|---|---|---|---|
| Intensity ($512 \times 512$ px$^2$) | 95 | 54.98 | 27.58 |
| Intensity ($100 \times 100$ px$^2$) | 6.2 | 70.41 | 33.49 |
| Landmark+PnP [Baseline] | 0.1 | 12.96 | 32.70 |
| Weighted Landmark+PnP [**Prop.**] | 0.9 | 2.73 | 6.97 |

Table 1: Comparison of intensity- and landmark-based methods, evaluated in terms of mean total registration time and median rotation and translation error. The Intensity method is the DiffDRR ((Gopalakrishnan and Golland, 2022)) projection metric with varying 2D image sizes, while Landmark+PnP use a U-Net landmark annotator fed into a direct pose optimization, either with or without our proposed weights (c.f. Finetune + Test Time CW in Table 2). The weights are estimated with MC dropout ($S = 100$).

depth-poor intra-operative fluoroscopy is complemented by registration to restore 3D context using standard operating room equipment (Cho et al., 2023).

Current methodology is divided into two broad classes of methods, image intensity matching (Unberath et al., 2018; Gao et al., 2020; Gopalakrishnan and Golland, 2022) and landmark matching (Gao et al., 2003; Lepetit et al., 2009; Li et al., 2012). The former uses a forward model of projection and iteratively updates beliefs about the detectors relative pose by matching that projection to the observed images. While this has the potential to have high accuracy and generality across anatomy, each forward pass is generally computationally expensive and thus often slow for bed-side applications. Landmark methods instead rely on anatomic knowledge of the target volume, and match pre-defined features or landmarks between the 2D and 3D sets. While this is much more computationally tractable, avoiding the reprojection steps of intensity matching, it is prone to higher errors due to sensitivity in point matching and the intrinsic variability of anatomical landmarks.

In the present work we propose an uncertainty-aware framework that models the reliability of each anatomical point during the landmark identification phase, and includes that estimate as an optimization weight in the subsequent pose estimation phase. By integrating per-landmark uncertainty into a fully differentiable landmark detection and Perspective-n-Point (PnP) pipeline, our method stabilizes pose estimation by increasing the influence of trustworthy keypoints and suppressing unreliable ones. Our contributions are threefold: (1) we introduce a differentiable uncertainty-to-weight formulation that enables continuous weighting during training and inference of Landmark-PnP pose estimation schemes; (2) we show that our selection strategies improve robustness even without requiring retraining; and (3) we provide empirical evidence that uncertainty estimates landmark reliability, yielding substantially improved 2D/3D pelvis registration performance.

## 2. Related Work

There are multiple existing approaches for rigid 2D/3D registration of radiography or fluoroscopy to volumetric CT. Most relevant to our work are landmark- and feature-based

methods, which assume correspondence between 3D anatomical landmarks in CT and their 2D projections (Bier et al., 2018; Grupp et al., 2020). These are the x-ray/fluoroscopy case of the Perspective-n-Point problem from general imaging (Gao et al., 2003; Lepetit et al., 2009; Li et al., 2012). We choose to solve this optimization using gradient based methods due to their relative simplicity and apparent quality for our domain.

Another broad class of 2D/3D registration methods are based on matching image intensity. These intensity-based methods align digitally reconstructed radiographs (DRRs) (Unberath et al., 2018; Gao et al., 2020; Gopalakrishnan and Golland, 2022) generated from CT with intra-operative fluoroscopy by optimizing image similarity measures such as correlation or information based metrics (Gopalakrishnan et al., 2024). These methods are general in the sense that they do not need outside knowledge about the content of the images, but scale poorly in the size of the images, both 2D and 3D, leading to long run times and heavy computational costs (see Table 1).

Uncertainty estimation in deep learning and uncertainty-aware architectures are relatively well studied. In this paper we use one of the early approaches, Monte Carlo (MC) dropout, where dropout layers (Srivastava et al., 2014) are kept active at test time and multiple stochastic forward passes are used to approximate epistemic uncertainty via the variance of the predictions (Gal and Ghahramani, 2016; Kendall and Gal, 2017). In medical image analysis, this idea has been applied to segmentation and landmark detection to highlight regions where the network is less confident and to guide human review or post processing (Jungo et al., 2018; Drevický and Kodym, 2020; Ye et al., 2023).

More complex uncertainty estimators are possible, but often do not fit our use criterion for downstream weighting. Ensemble methods (Rahaman et al., 2021) have been proposed as a Dropout generalization, as each Dropout iteration is often viewed as an ad hoc bootstrapped ensemble, but these require retraining and multiple network evaluations (beyond randomly sampled masks), and so they do not fit our use-case. Another family of methods is the conformal prediction (Shafer and Vovk, 2008) framework, where "conformance scores" effectively rate datapoints as out-liers or in-liers, allowing classification or regression to split its operating characteristic curves into a geometric product. Our method is not a direct classification, and conformance scores are only related to uncertainty by quantile/order; there is no guarantee of magnitude differences being related. Bayesian methods may also model and then sample parameter weights to form posterior distributions of both networks and outputs (Marinescu et al., 2020), but these sampling methods are often slow, and again require numerous network evaluations.

## 3. Method

### 3.1. Problem Setup and Groundwork

We would like to compute the pose of a patient observed under 2D fluoroscopy or radiograph relative to a standardized 3D pose, for a fixed detector (x-ray camera) position; this is equivalent to finding the pose of a detector which is observing a patient in standard position. The search space is over all 3D pose parameters $\theta = (r, t)$, composed of a rotation $r$ and a translation $t$. This generally has 6 degrees of freedom, 3 dof for $r$ and $t$ each, even though $r$ may be represented in overparameterized fashion, i.e., by a $3 \times 3$ matrix or a 4 dimensional quaternion.

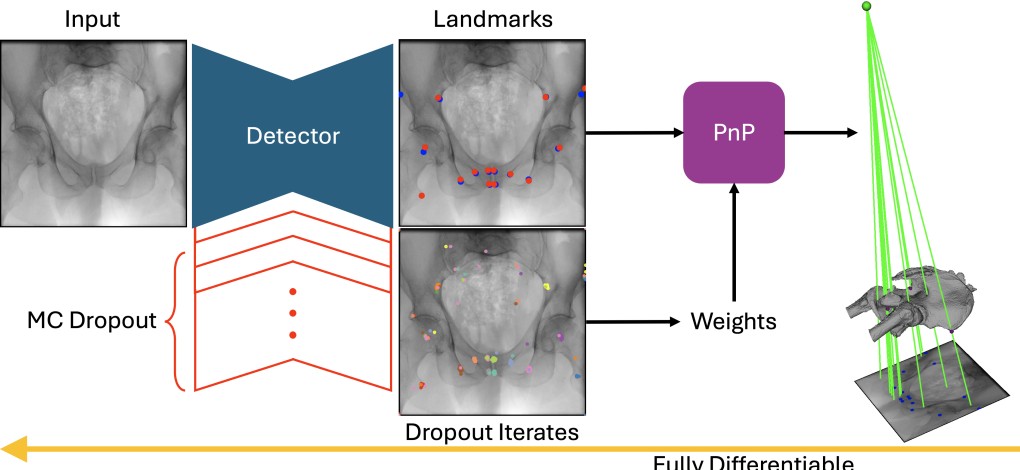

Figure 1: Overview of the uncertainty-aware pose estimation framework. Monte Carlo (MC) Dropout is used to produce uncertainty estimates (in Red, below Detector) from the primary landmarking network (in Blue, Detector), which then weight landmarks during the Perspective-n-Point (PnP) optimization (in Purple). The PnP method has no learnable parameters, and is fully differentiable, allowing registration losses to be propagated directly back to the landmarking network.

For general target volumes and their corresponding 2D images, image matching methods can be employed to efficiently search this space (Grupp et al., 2020; Gopalakrishnan et al., 2025), but these methods usually disregard any special knowledge of the target image domain (continued discussion in Section 6 and (Gao et al., 2020)). In surgical imaging we often know much more about the target region's anatomy and potential keypoints/features. Assuming we have corresponding features, we can avoid the image matching problem and instead solve a 2D/3D point-set registration. Using a fixed detector convention, we write the point correspondence problem as the following least squares problem between landmarks in 3D ($p^{3D}$) and their apparent 2D positions in the image ($p^{2D}$):

$$\min_{\theta} \sum_{i}^{L} \| \operatorname{Proj}[T_\theta(p_i^{3D})] - p_i^{2D} \|_2^2. \quad \text{PnP Least Squares Problem} \tag{1}$$

Here, Proj is the operator that takes 3D points to their 2D positions in the camera. The least squares 2D/3D point alignment problem has been solved in the literature by the Perspective-n-Point (PnP) family of methods (Lepetit et al., 2009; Terzakis and Lourakis, 2020). We use a gradient based optimization which in practice converges to the correct solution. Notably this optimization is itself fully differentiable (Amos and Kolter, 2017), as are, generally speaking, most of the class of PnP solvers.

For landmark prediction, we employ a U-Net-based convolutional neural network to generate landmark probability maps from fluoroscopy images (Mika et al., 2025), as shown in Figure 1. During inference, the coordinate for each landmark is identified as the pixel location with the maximum intensity in the corresponding predicted heatmap. For image

inputs $I$ and neural network $f$ parameterized by $\phi$, we write this operation as:

$$p_i^{2\mathrm{D}} = \operatorname*{argmax}_{x \in \Omega} \left( [f(I; \phi)]_i \right). \tag{2}$$

Here $\Omega$ is the 2D image domain in which point $x$ is contained. The $i^{th}$ channel output corresponds to the $i^{th}$ landmark.

Our specific architecture incorporates a ResNet-101 encoder (He et al., 2016) initialized with ImageNet-pretrained weights (Deng et al., 2009). To improve generalization, we utilize a dilation-erosion label augmentation scheme, which has demonstrated efficacy in orthopaedic datasets by broadening the effective training signal (Suh et al., 2023; Chan et al., 2025). The model is trained to predict these augmented heatmaps using a binary cross-entropy loss. As demonstrated in (Mo et al., 2025), because the PnP operation is differentiable, we can add a weighted PnP loss directly to the binary cross-entropy loss to improve landmark identification.

## 3.2. Main Contribution: Uncertainty Weighted PnP and PnP Losses

Similar to other least squares estimation problems, PnP methods are notoriously susceptible to outliers; one solution to this general problem is uncertainty weighted least squares solution (Hastie, 2009). We introduce this same solution for our PnP solver. Modifying Eq. 1, we construct a weighted least squares PnP problem by adding weights $w_i$ to each of the point error terms:

$$\min_{\theta} \sum_{i}^{L} w_i \| \operatorname{Proj}[T_{\theta}(p_i^{3\mathrm{D}})] - p_i^{2\mathrm{D}} \|_2^2. \quad \text{PnP Weighted Least Squares Problem} \tag{3}$$

The $w_i$ should be set to values that are proportional to the "trustworthiness" of each point, or equivalently inversely proportional to the uncertainty for each point.

As the points are selected by neural network, it is thus natural to use a neural network uncertainty method to estimate this quantity. We estimate $w_i$ using MC Dropout uncertainty (Gal and Ghahramani, 2016), which prescribes Monte Carlo sampling for each of the outputs with stochasticity added via Dropout on the estimating neural network, then measuring uncertainty by summary statistics on those samples; in our context, this means we should compute $p_{i,s}$ using

$$p_{i,s}^{2\mathrm{D}} = \operatorname*{argmax}_{x \in \Omega} \left( [f(I, m_s \circ \phi)]_i \right). \tag{4}$$

where $m_s$ is the Dropout mask for sample $s$ (Kendall and Gal, 2017). We do this a total of $S$ times for different $m_s$ masks; for PyTorch implementations, these masks are efficiently sampled in memory, so that batched outputs see significant parallelism gains, computing all $S$ samples in parallel, or as many as memory allows.

We then compute summary statistics $\bar{p}_i$ and $u_i$ for each of the $i$ landmarks:

$$\bar{p}_i = \frac{1}{S} \sum_{s=1}^{S} p_{i,s}^{2\mathrm{D}} \qquad u_i = \sqrt{\frac{1}{S} \sum_{s=1}^{S} \| p_{i,s}^{2\mathrm{D}} - \bar{p}_i \|_2^2}. \tag{5}$$

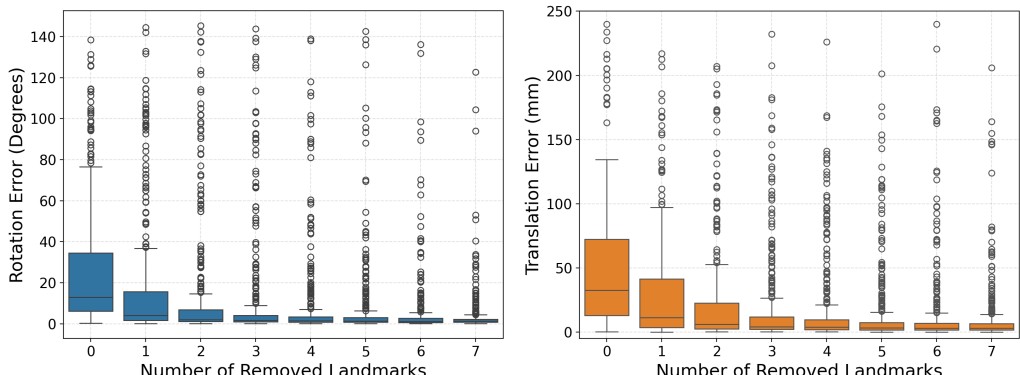

Figure 2: Oracle experiment using ground-truth 2D landmark positions. The boxplots show the rotation error and translation error as a function of number of removed landmarks. The boxplots from left to right correspond to oracle landmark filtering levels $K = 0, 1, \ldots, 7$, i.e., removing the $K$ most erroneous landmarks.

Statistical theory (Kendall et al., 2018) suggests that our weights should be inversely proportional to their uncertainty. Whether due to inaccuracy while computing the weights, or due to other deviations, we find that it is more numerically stable to normalize and then negatively exponentiate the weights:

$$\tilde{u}_i = \frac{u_i}{\max_{i'} u_{i'} + \varepsilon} \qquad w_i = \exp(-\beta \tilde{u}_i), \qquad (6)$$

with $\beta$ as a hyper-parameter controlling weight "fall-off", which will correspond to outlier suppression strength in the resulting optimization of Eq. 3. For numerical stability of the optimization we also normalize $w_i$ after this procedure. We refer to solutions of Eq. 3 as *continuous weighting*.

**Implementation considerations:** We can implement this scheme directly in Pytorch *both for inference and training*. The Dropout statistics themselves are composed of fully differentiable operations, as are the weight constructions and the weighted PnP optimization. However, for completely untrained networks, these weights will likely be nearly uniform (i.e., not informative). Thus, we implement using a finetuning scheme, where a network is trained to output landmarks first, before being refined by weighted PnP losses.

In backpropagating from the weighted PnP to the primary network, we need to propagate through all of our Dropout iterates. This requires a number of network activations to be held in memory that is equal to the dropout iterates; to avoid this cost, we could choose to exclude the dropout from the backpropagation. This would lead to inaccuracies in the gradient, but as we show in Table 2, this only leads to small overall performance degradation for significant memory overhead reduction.

**Test-time filtering and discrete selection:** We find that instead of performing weighted squares, wholly excluding low weight landmarks from the optimization empirically produces strong performance. Ranking landmarks by uncertainty, we define an uncertainty-filtered subset by discarding the $K$ most uncertain visible landmarks. This method we

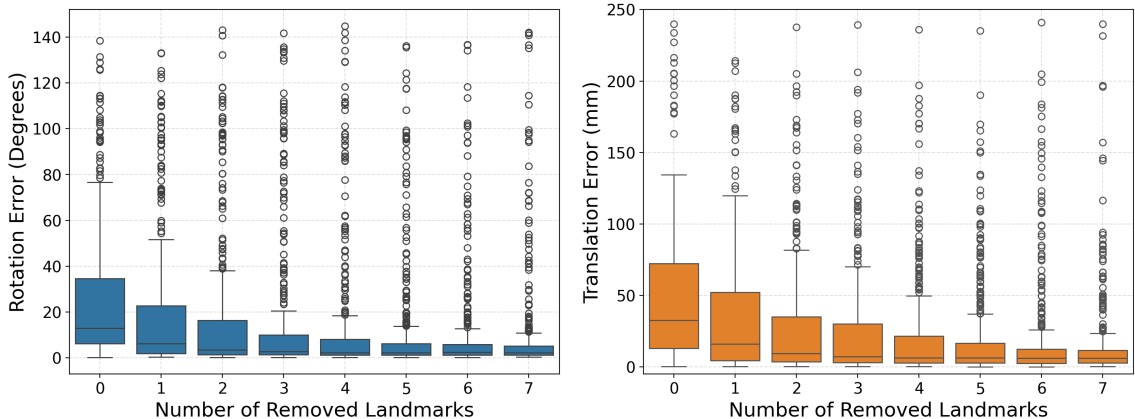

Figure 3: Rotation and translation errors across landmark dropout iterations ($K = 0, \ldots, 7$). Each boxplot summarizes the per-image error for a given dropout number $K$ after aggregating results across all patients. The translation axis is truncated at 250 mm for outliers exceeding this threshold in $K = 0, 1, 2, 3, 5$ (1.39%), $K = 4$ (1.11%), $K = 6$ (0.83%), and $K = 7$ (0.28%).

call *discrete selection*; while it is not easily optimized over, at test time it provides good performance as shown in Section 5.

## 4. Experiments

We validated our approach using the DeepFluoro dataset (Grupp et al., 2020), which provides pelvic CT volumes, paired 3D anatomical landmarks, fluoroscopy images, and its necessary pose parameters (see Appendix A). Using the CT volumes, we generated DRRs via DiffDRR (Gopalakrishnan and Golland, 2022) for the synthetic experiments, and we additionally evaluated the method on the fluoroscopy images provided in the same dataset. The imaging geometry was standardized with a source-to-detector distance of 1020 mm and a volume-to-detector distance of 400 mm. To simulate diverse patient positioning, we randomized camera poses (pelvic poses) with rotations drawn from $[-45°, 45°]$ along the $x$- and $y$-axes and $[-15°, 15°]$ along the $z$-axis. Translations were sampled independently from $[-50, 50]$ mm along each axis. For every rendered DRR ($512 \times 512$), the 14 3D landmarks were projected to generate ground-truth 2D labels, forming the basis for detection and registration tasks.

We employed a leave-one-subject-out cross-validation strategy, holding out one volumetric image and all associated 2D images as a test set while training and fine-tuning on the remaining subjects. To quantify uncertainty, we incorporated MC dropout ($p = 0.1$) within the decoder. To estimate the uncertainty, the model was evaluated $S$ times per image ($S = 40$ for fine-tuning, $S = 100$ for testing) to generate a distribution of sample predictions $\{p_{i,s}\}_{s=1}^{S}$ for each landmark $i$.

To assess the benefit of removing erroneous landmarks, we conducted an oracle experiment (Figure 2), where pose accuracy was measured by the Euler angle difference for

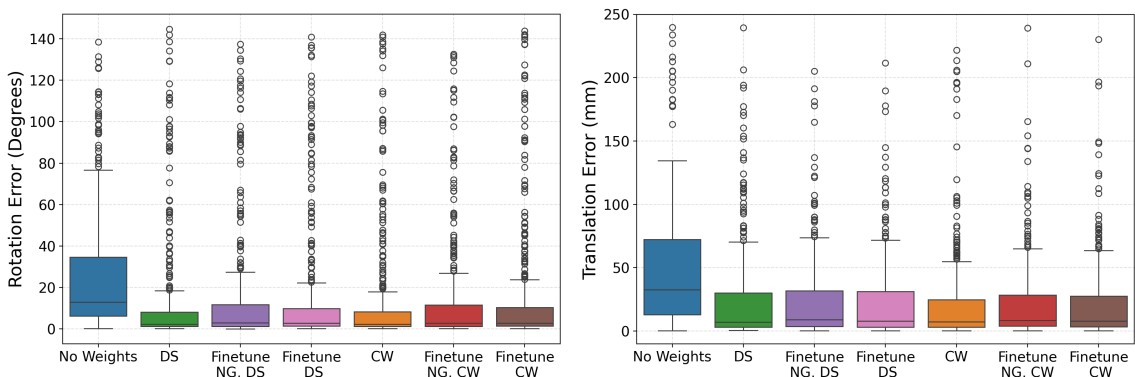

Figure 4: Rotation and translation error distributions comparing No Weights, Discrete Selection (DS, top-3 landmark filtering) and Continuous Weighting (CW). Variants include finetuning without gradient updates on MC dropout model (Finetune NG) and fully finetuned model (Finetune). The translation axis is truncated at 250 mm for outliers. In the translation boxplot, from left to right, 1.11%, 0.83%, 0.28%, 0.28%, 0.83%, 0.28%, 0.28% of the datapoints were truncated.

rotation and by translation error (mm), computed as the Euclidean norm between the predicted and ground-truth translation vectors. For each landmark $i$, we compute the oracle detection error as $d_i = \|p_i^{2D} - p_i^*\|_2$ and filter out the top-$K$ highest-error landmarks to form an oracle filter $w_i^{gt}$. Excluding these landmarks consistently improved both rotation and translation accuracy, motivating our use of uncertainty $u_i$ as a proxy for the unknown error $d_i$.

For the synthetic experiments on CT-derived DRRs, we evaluated pose estimation performance using the same rotation and translation metrics. To assess our framework, we designed four experimental stages. We begin by quantifying registration stability by applying uncertainty-based top-$K$ filtering ($K = 0, \ldots, 7$). Next, we compare our uncertainty-weight-based fine-tuned model against a baseline that treats all landmarks equally, as well as test-time-only weighting methods. We then analyze per-landmark uncertainty to show which anatomical regions benefit from uncertainty-aware handling. Finally, we perform an error retention analysis by progressively excluding high-uncertainty images to validate the effectiveness of the metric as an outlier detector.

Additionally, we evaluated performance on the fluoroscopy images. One specimen was used for testing, and the remaining five specimens were used for training and validation. For landmark detection, the network was first trained on synthetic DRRs and then fine-tuned on fluoroscopy images to better adapt to the appearance of clinical acquisitions. Fluoroscopy-image-based pose estimation was performed using the native DeepFluoro calibration and pose convention (see Appendix A). In addition to the same rotation and translation metrics, we report mean target registration error (mTRE), defined as the mean Euclidean distance between the 3D landmarks transformed by the estimated pose and by the ground-truth pose in the native DeepFluoro geometry.

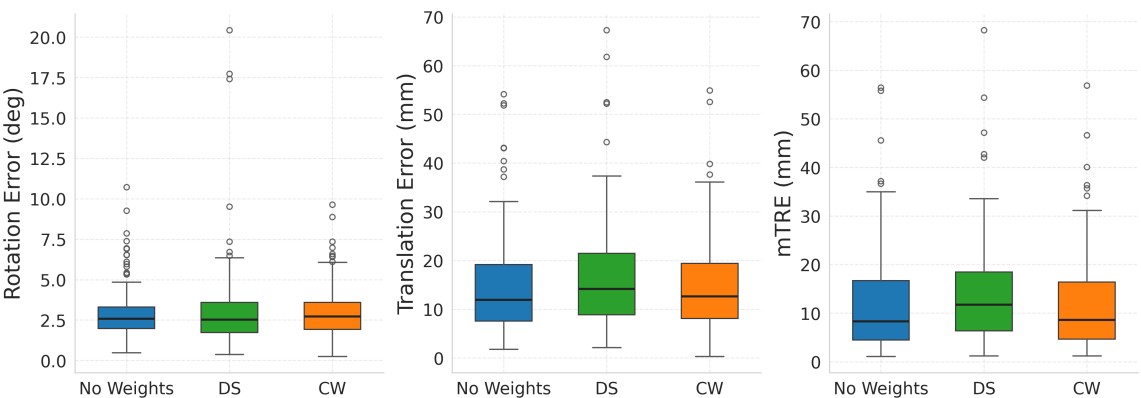

Figure 5: DeepFluoro fluoroscopy image registration performance. Left to right: rotation error, translation error, and mTRE for *No Weights*, discrete selection (DS, $K = 3$), and continuous weighting (CW). Boxplots exclude gross failures with rotation error $> 20°$, translation error $> 70$ mm, or mTRE $> 70$ mm for readability.

## 5. Results

We evaluated the efficacy of the estimated uncertainty as a criterion for outlier rejection, independent of ground-truth labels. Figure 3 illustrates the distributions of rotation and translation errors as we progressively exclude out the top-$K$ most uncertain landmarks ($K = 0, \ldots, 7$). We observed a sharp, monotonic decay in both error and interquartile range (IQR) as $K$ increases. This suggests that the uncertainty metric $u_i$ effectively isolates the long-tail outliers that disproportionately destabilize the registration solver. By removing these high-uncertainty points, the system recovers a geometrically consistent subset of landmarks, resulting in precise pose estimation even in the presence of detection noise.

Figure 4 and Table 2 summarize 3D pose estimation performance across seven experimental configurations. The unweighted baseline that uses all landmarks equally during pose estimation shows high variance with frequent outliers, resulting in a mean rotation error of 26.14 degrees and a mean translation error of 51.18 mm. Introducing uncertainty-based discrete selection (DS) at inference nearly halves both errors (14.22 degrees, 24.79 mm), while fine-tuning with continuous weighting (CW) maintains similar performance with lower translation error (21.92 mm). CW applied directly at inference achieves the lowest mean rotation error (13.94 degrees), while the combined strategy with fine-tuning yields the best overall translation accuracy (20.63 mm).

On fluoroscopy images, uncertainty-aware continuous weighting produced the strongest overall registration performance. As shown in Figure 5, the filtered boxplot show that CW achieved the lowest translation error and the lowest mTRE among the three methods, while maintaining rotation accuracy comparable to the unweighted baseline. Quantitatively, *No Weights*, DS, and CW achieved median rotation errors of 3.12, 5.99, and 3.54 degrees, median translation errors of 18.69 mm, 32.61 mm, and 18.17 mm, and median mTRE values of 16.09 mm, 29.91 mm, and 15.80 mm, respectively. In contrast to the synthetic setting, DS performed worse on real fluoroscopy, likely because many images contain only

| Experiment | Rotation Error | P=50 | P=60 | P=70 | P=80 | P=90 |
|---|---|---|---|---|---|---|
| No Weights | 26.14 ± 30.08 | 12.96 | 17.45 | 29.92 | 41.46 | 75.48 |
| DS | 14.22 ± 29.18 | 2.31 | 3.39 | 5.42 | 13.68 | 51.83 |
| Finetune + NG + DS | 16.52 ± 30.38 | 2.87 | 4.51 | 9.24 | 21.09 | 57.17 |
| Finetune + DS | 16.33 ± 30.78 | 2.84 | 4.37 | 8.08 | 20.00 | 59.39 |
| CW | 13.94 ± 28.67 | 2.27 | 3.43 | 5.36 | 12.72 | 46.68 |
| Finetune + NG + CW | 15.65 ± 28.57 | 2.73 | 4.45 | 9.23 | 20.65 | 54.13 |
| Finetune + CW | 15.84 ± 30.80 | 2.73 | 4.78 | 7.41 | 16.97 | 49.39 |

| Experiment | Translation Error | P=50 | P=60 | P=70 | P=80 | P=90 |
|---|---|---|---|---|---|---|
| No Weights | 51.18 ± 56.58 | 32.70 | 45.09 | 64.30 | 80.86 | 111.67 |
| DS | 24.79 ± 50.58 | 6.38 | 9.09 | 13.59 | 33.46 | 70.17 |
| Finetune + NG + DS | 21.98 ± 33.38 | 7.46 | 11.21 | 21.61 | 33.64 | 60.30 |
| Finetune + DS | 21.92 ± 36.44 | 7.21 | 10.53 | 17.19 | 32.08 | 55.74 |
| CW | 24.18 ± 62.23 | 6.35 | 8.98 | 12.96 | 27.99 | 58.35 |
| Finetune + NG + CW | 22.14 ± 34.78 | 7.44 | 10.78 | 20.02 | 32.67 | 58.41 |
| Finetune + CW | 20.63 ± 32.54 | 6.97 | 10.24 | 18.84 | 30.99 | 55.14 |

Table 2: Quantitative analysis of pelvic pose estimation comparing No Weights, Discrete Selection (DS, top-3 landmark filtering) and Continuous Weighting (CW). Variants include finetuning without gradient updates on MC dropout model (Finetune NG) and fully finetuned model (Finetune). We report the mean ± standard deviation together with the 50th, 60th, 70th, 80th, and 90th percentiles of the rotation error (degrees) and translation error (mm)

a limited number of visible landmarks, so removing three uncertain landmarks can leave an insufficient subset for stable registration.

Figure 6 presents error retention curves showing a monotonic reduction in residual error as samples with high uncertainty, defined as the mean spatial deviation across all the landmarks in the image, are progressively excluded. In intra-operative guidance, this facilitates graceful failure by allowing the system to withhold prediction on ambiguous frames rather than outputting misleading guidance. Therefore, clinical workflows can strategically trade off temporal for reliability to ensure that surgical decision making is informed exclusively by high-confidence pose estimates.

## 6. Discussion

Our results demonstrate that using estimates of epistemic uncertainty in landmark estimation improves downstream registration accuracy. In synthetic experiments, both CW and DS improve robustness, whereas in fluoroscopy experiments, CW is more reliable than hard landmark removal.

Because anatomical landmarks often represent abstract geometric constructs from physical anatomy rather than visually distinct image features, human annotation can contain

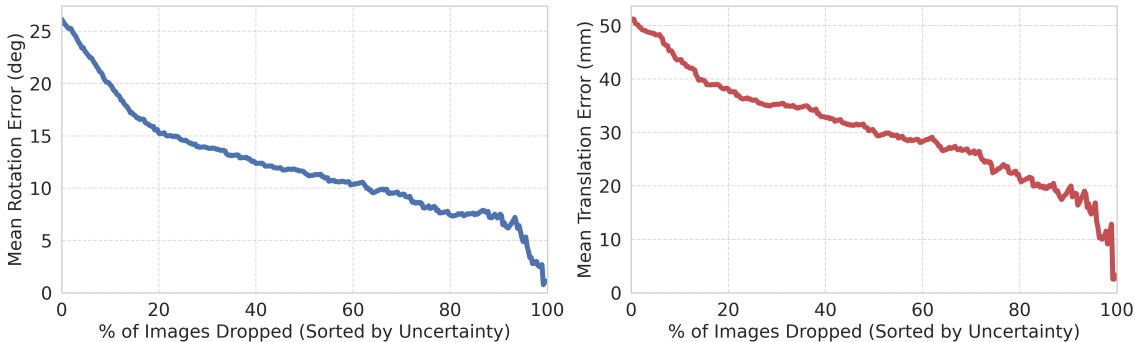

Figure 6: Mean error as samples are removed in order of their estimated uncertainty. The curves show the reduction in mean rotation (left) and translation (right) errors as the most uncertain samples are progressively filtered out (0% to 100%).

subjective variability. Accordingly, although our MC-dropout procedure is used as a practical estimator of epistemic uncertainty, the resulting uncertainty scores may also reflect ambiguity in landmark visibility or annotation consistency. From the standpoint of pose estimation, however, this remains useful: landmarks that are uncertain for either reason are precisely those that should contribute less to the downstream registration objective. However, the current mechanism for weighting and filtering relies on fixed hyperparameters ($\beta$ and $K$), which do not account for the significant variance in clinical image quality. A more optimal future approach might adapt $\beta$ and $K$ dynamically, although such methods may themselves become unstable.

## 7. Conclusion

We present an uncertainty-aware framework for landmark-based 2D/3D pelvis registration that integrates epistemic uncertainty into both continuous weighting and discrete landmark selection. By estimating per-landmark reliability with MC dropout and incorporating these estimates into the PnP formulation, our method reduces sensitivity to erroneous detections and improves registration accuracy compared to conventional landmark-based registration. Across synthetic and fluoroscopy image evaluations, continuous weighting provided the most consistent gains, while discrete selection was beneficial primarily in the synthetic setting. Overall, these results demonstrate that uncertainty provides a principled mechanism for identifying unreliable keypoints, stabilizing pose estimation, and enabling graceful failure on ambiguous frames, with continuous weighting offering the most consistent improvements across both simulated and real fluoroscopic data.

## Acknowledgments

This work was supported in part by NSF 2321684, NIH Training Grant T32 EB001628-20, and a VISE Seed Grant.

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

## Appendix A. Data Collection and Preprocessing

### A.1. CT Volume and Segmentation Reorientation

The DeepFluoro dataset (Grupp et al., 2020) provides each CT volume with voxel array $V$, voxel spacing $s = (s_x, s_y, s_z)$, direction matrix $R \in \mathbb{R}^{3 \times 3}$, and physical origin $o \in \mathbb{R}^3$. The scanner index–to–world affine matrix is

$$A = \begin{bmatrix} R \operatorname{diag}(s_x, s_y, s_z) & o \\ 0 & 1 \end{bmatrix}. \tag{7}$$

To match the NIfTI $(x, y, z)$ axis convention, we apply a permutation of voxel indices followed by a reflection. Let $V(i, j, k)$ denote the original voxel array with axes $(i, j, k)$. First, we permute axes 0 and 2 via

$$\tilde{V}(i, j, k) = V(k, j, i). \tag{8}$$

Next, we flip the new $x$–axis (dimension $i$) by reflecting it:

$$V'(i, j, k) = \tilde{V}(N'_x - 1 - i, \; j, \; k), \tag{9}$$

where $N'_x$ is the size of the permuted first dimension. Together, this yields the reoriented volume $V'$ used for all NIfTI exports. This flip corresponds to the matrix

$$F = \begin{bmatrix} -1 & 0 & 0 & (N_z - 1)s_z \\ 0 & 1 & 0 & 0 \\ 0 & 0 & 1 & 0 \\ 0 & 0 & 0 & 1 \end{bmatrix}, \tag{10}$$

so the final NIfTI affine is

$$A' = AF. \tag{11}$$

This guarantees that $(V', A')$ preserves the original scanner physical coordinates.

### A.2. 3D Landmark Conversion

Each anatomical landmark is provided in physical scanner coordinates $p = (x, y, z)^\top$. To map it into the voxel space of the reoriented CT, we apply

$$v = (A')^{-1} \begin{bmatrix} p \\ 1 \end{bmatrix}, \qquad v = (i, j, k). \tag{12}$$

The resulting voxel coordinates are rounded to the nearest integer index.

### A.3. 2D Projection Extraction

Projection images are intensity-normalized to $[0, 255]$ and saved as PNG files. The provided 2D landmark coordinates $(u, v)$ are written directly unless they lie outside the image bounds, in which case the corresponding landmark is marked as invisible for the ground truth landmarks.

### A.4. Fluoroscopy Image Pose Evaluation in Native DeepFluoro Geometry

For the fluoroscopy image experiments, pose estimation was performed directly in the native DeepFluoro HDF5 geometry rather than in the standardized synthetic DRR geometry (Suh et al., 2025). For each projection, the dataset provides 3D anatomical landmarks in the pelvis-volume coordinate system, corresponding 2D detector annotations, the pelvis pose `cam-to-pelvis-vol`, and the global projection calibration matrices `intrinsic` and `extrinsic`. Let $\mathbf{X}_i \in \mathbb{R}^3$ denote a 3D landmark in pelvis-volume coordinates and let

$$T_{\text{cam}\to\text{pelvis}} \in \mathbb{R}^{4\times4} \tag{13}$$

denote the stored HDF5 pose. The corresponding world-to-camera transform used for projection was defined as

$$T_{w2c} = E\, T_{\text{cam}\to\text{pelvis}}^{-1}, \tag{14}$$

where $E \in \mathbb{R}^{4\times4}$ is the dataset extrinsic matrix. The projection matrix was then

$$P = K\,[T_{w2c}]_{1:3,:}, \tag{15}$$

with $K \in \mathbb{R}^{3\times3}$ the HDF5 intrinsic matrix. Writing a landmark in homogeneous form as

$$\tilde{\mathbf{X}}_i = \begin{bmatrix} \mathbf{X}_i \\ 1 \end{bmatrix}, \tag{16}$$

its projected detector coordinate is

$$\tilde{\mathbf{x}}_i = P\tilde{\mathbf{X}}_i = \begin{bmatrix} \hat{u}_i \\ \hat{v}_i \\ \hat{w}_i \end{bmatrix}, \qquad \mathbf{x}_i = \begin{bmatrix} \hat{u}_i/\hat{w}_i \\ \hat{v}_i/\hat{w}_i \end{bmatrix}, \tag{17}$$

which yields the 2D landmark in the native DeepFluoro detector pixel frame.

The landmark detector predicts coordinates in the resized model image, after which the predictions are mapped back to native detector resolution before registration. If $(u_i^m, v_i^m)$ denotes a predicted point in the model image of size $W_m \times H_m$, and $(W, H)$ denotes the raw detector size, then the corresponding detector-space coordinate is computed as

$$u_i = u_i^m \frac{W}{W_m}, \qquad v_i = v_i^m \frac{H}{H_m}. \tag{18}$$

These detector-space landmarks are then paired with the HDF5 3D landmarks for pose recovery.

Unlike the original synthetic registration path, this fluoroscopy image evaluation uses a new raw-geometry solver matched to the HDF5 convention. Pose was initialized from the visible 2D–3D correspondences using DLT and then refined by Levenberg–Marquardt nonlinear least squares. The resulting estimate is

$$\hat{T}_{w2c} = \arg\min_T \sum_{i=1}^{L} \|\pi(\mathbf{X}_i; T) - \mathbf{x}_i\|_2^2, \tag{19}$$

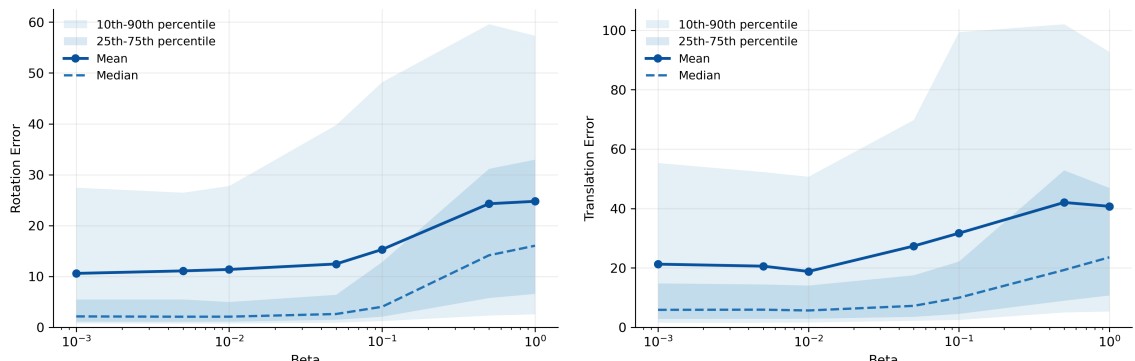

Figure 7: Sensitivity of continuous weighting to the fall-off parameter $\beta$. Left: rotation error. Right: translation error. Solid lines denote the mean and median across all held-out test images, while shaded regions denote the 25th–75th and 10th–90th percentiles. Performance is best in the low-$\beta$ regime (0.001–0.01), while larger $\beta$ values increase both the central error and the upper tail of the error distribution.

where $\pi(\mathbf{X}_i; T)$ denotes the projected detector coordinate under transform $T$, and $L$ is the number of visible correspondences used for that image.

For the uncertainty-aware continuous-weighting variant, each landmark is assigned a weight $w_i$ derived from MC-dropout variability, and the weighted objective is

$$\hat{T}_{w2c}^{\mathrm{CW}} = \arg\min_T \sum_{i=1}^{L} w_i \left\| \pi(\mathbf{X}_i; T) - \mathbf{x}_i \right\|_2^2. \tag{20}$$

The fluoroscopy image evaluation follows the same landmark-driven registration principle as the synthetic experiments, but is carried out directly in the native DeepFluoro calibration and pose convention.

To evaluate pose accuracy in the native DeepFluoro geometry, we additionally report mean target registration error (mTRE). Let $T_{w2c}^*$ denote the ground-truth world-to-camera transform and $\hat{T}_{w2c}$ the estimated transform. Then mTRE is computed over the same 3D landmark set as

$$\mathrm{mTRE}(T_{w2c}^*, \hat{T}_{w2c}) = \frac{1}{L} \sum_{i=1}^{L} \left\| T_{w2c}^* \tilde{\mathbf{X}}_i - \hat{T}_{w2c} \tilde{\mathbf{X}}_i \right\|_2, \tag{21}$$

where $L$ is the number of landmarks used for evaluation and $\tilde{\mathbf{X}}_i$ denotes the homogeneous form of the $i$-th 3D landmark. Thus, mTRE measures the mean Euclidean discrepancy between landmark positions under the estimated and ground-truth poses in camera space.

## Appendix B. Sensitivity to Hyperparameter $\beta$

Continuous weighting is defined by converting normalized landmark uncertainty $\tilde{u}_i$ into reliability weights via

$$w_i = \exp(-\beta \tilde{u}_i),$$

where $\beta$ controls the rate at which uncertain landmarks are suppressed. To assess the robustness of our method to this design choice, we performed a sensitivity analysis over $\beta \in \{0.001, 0.005, 0.01, 0.05, 0.1, 0.5, 1.0\}$ while keeping all other settings fixed.

Figure 7 summarizes the resulting rotation and translation error distributions across all held-out test images. For each $\beta$, we report the mean and median together with the 25th–75th and 10th–90th percentile bands in order to characterize both central tendency and spread.

The results show that performance is best and most stable in the low-$\beta$ regime (0.001–0.01). In this range, both rotation and translation errors remain low across the median and upper percentiles. As $\beta$ increases, both the central error and the spread of the distribution grow, with especially clear degradation for $\beta \geq 0.1$. This trend indicates that overly aggressive fall-off suppresses not only unreliable landmarks, but also landmarks that still provide useful geometric information for pose estimation.

