# OpenReview forum: "Landmark Detection Uncertainty as a Reliability Weight for Robust Landmark-based 2D/3D Pelvic Pose Estimation"
_MIDL.io/2026/Conference — MIDL 2026 Poster_

### Official Review · Reviewer_igPY · 2026-01-08

**Confidence:** 4
**Preliminary Rating:** 3

**Summary:**

This paper proposes an uncertainty-aware landmark-based 2D/3D pelvic pose estimation framework that explicitly models epistemic uncertainty in fluoroscopic landmark detection and incorporates it into the Perspective-n-Point (PnP) formulation. Using Monte Carlo dropout, the method estimates per-landmark uncertainty and leverages it either through continuous weighting in a weighted PnP optimization or discrete selection by removing highly uncertain landmarks at inference time. Experiments on synthetic fluoroscopy generated from a public pelvic CT dataset demonstrate that uncertainty-guided weighting and filtering substantially improve registration robustness and accuracy compared to standard landmark-based approaches

**Strengths:**

1 The paper provides a way to translate epistemic uncertainty into reliability weights, significantly stabilizing pose estimation in the presence of noisy or ambiguous landmarks
2 Both discrete selection and continuous weighting improve rotation and translation accuracy by large margins over the baseline, without requiring new detectors or expensive intensity-based registration
3 The uncertainty-weighted PnP formulation is differentiable end-to-end and is highly desirable for image-guided surgical workflows

**Weaknesses:**

1 It’s 2026; the topic and the method itself are not so exciting.

2 The uncertainty-to-weight mapping and the number of discarded landmarks are manually set and not adaptive.

3 Only synthetic data are used in the experiments.

**Detailed Comments:**

Refer to the strengths and weaknesses section.

**Justification Of The Preliminary Rating:**

This paper proposes an uncertainty-aware landmark-based 2D/3D pelvic pose estimation framework that explicitly models epistemic uncertainty in fluoroscopic landmark detection and incorporates it into the Perspective-n-Point (PnP) formulation. Using Monte Carlo dropout, the method estimates per-landmark uncertainty and leverages it either through continuous weighting in a weighted PnP optimization or discrete selection by removing highly uncertain landmarks at inference time. Experiments on synthetic fluoroscopy generated from a public pelvic CT dataset demonstrate that uncertainty-guided weighting and filtering substantially improve registration robustness and accuracy compared to standard landmark-based approaches

It's an OK work.

**Questions To Address In The Rebuttal:**

Not much. Just an average work.

---

> ### Author Response · Authors · 2026-01-24
> **Response for Reviewer igPY**
>
> We thank you for your time and constructive feedback.
>
> ---
> **Reviewer** `igPY` questioned the novelty of the proposed method.
> **Our Reponse**:
> We respectfully submit that the value of this work lies in solving the critical clinical challenge of registration instability with a straightforward and interpretable mechanism. Our approach provides a transparent solution that effectively isolates unreliable landmarks, yielding significant performance gains without unnecessary algorithmic complexity. Moreover, such a method was previously unexplored for our specific application. We believe this combination of interpretability, robustness, and application novelty is highly significant and holds practical clinical impact. We plan to extend this framework to orthopedic surgical workflows in the operating room which requires robust and interpretable fluoroscopy-based registration.
>
> ---
> **Reviewer** `igPY` aised concerns about manually set hyperparameters for uncertainty weighting and landmark removal.
> **Our Reponse**:
> The threshold K was determined empirically using a separate validation set to maximize the trade-off between outlier rejection and pose estimation accuracy. We agree that automatic hyper-parameter estimation would be useful, but we also believe that these methods will largely rely on the same validation-holdout schema to set those parameters. Thus, derivation of an analytic criterion seems very useful but difficult and potentially fragile (similar to selecting the number of clusters in a clustering method, or automatic relevance determination [ARD]). If we find that an automatic threshold determination method is feasible and performant, we will validate it in future work.
>
> ---
> **Reviewer** `igPY` noted that the experimental evaluation is limited to synthetic data.
> **Our Reponse**:
> We fully agree that validation on real clinical or cadaveric data is the ultimate standard for ensuring translational utility. However, as a large-scale dataset with paired intra-operative fluoroscopy and accurate ground-truth 3D pose was not available for this work, we utilized high-fidelity synthetic data as a necessary first step to establish the methodological validity of our uncertainty framework.

---

### Official Review · Reviewer_Bdjs · 2026-01-10

**Confidence:** 4
**Preliminary Rating:** 4
**Final Rating:** 4

**Summary:**

This work proposes a landmark-based framework for 2D/3D pelvis registration. The method incorporates uncertainty modeling for each landmark to improve the robustness and accuracy of the landmark prediction stage while maintaining a fully trainable end-to-end pipeline. The proposed approach is evaluated using synthetic fluoroscopy images generated from a public pelvis CT dataset. Experimental results show that the method improves landmark selection and consequently stabilizes pose estimation in the 2D–3D pelvis registration task.

**Strengths:**

1) The paper is generally well written, with the introduction and related work sections adequately covering the literature on 2D/3D registration, and the implementation section explained in sufficient detail, making the ideas accessible to a broader audience from diverse backgrounds.

2) The research motivation is well justified, as X-ray/fluoroscopy to CT registration is an essential task in many image-guided intervention workflows.

3) The proposed methodology is technically sound. While individual components, such as Monte Carlo dropout for uncertainty estimation and weighted least squares optimization, are based on established techniques, their integration into the overall pipeline is well reasoned. Applying them wisely in the keypoint detection phase within the 2D/3D registration makes it incremental but sufficiently novel for MIDL.

**Weaknesses:**

1) As discussed above, the utility of existing techniques (MC Dropout for uncertainty estimation, weighted least squares for outlier suppression, etc..) potentially limits the novelty of the proposed method. It might be interesting to explore connections to learning-based keypoint discovery methods, such as the work by Ruhaak et al. (10.1109/TMI.2017.2691259), which propose automatically learned geometric keypoints for registration. Integrating such approaches with uncertainty-aware weighting could represent a promising direction for future work.

2) It is not clear how the fixed value of K (the landmark removal threshold) is determined in the discrete selection step. Moreover, although this limitation is briefly acknowledged in the Discussion section, the reliance on static hyperparameters may significantly limit the generalization and practical applicability of the proposed method, particularly under varying image quality and clinical acquisition conditions.

3) The proposed method is solely evaluated on synthetic DDR data, which might lack realistic noise and anatomical variability. It is unclear whether the uncertainty estimates would remain meaningful under real intra-operative imaging conditions.

4) In the experiment section, the paper compares primarily against unweighted PnP and intensity-based registration, but does not include comparisons to classical pose estimation techniques such as RANSAC-based PnP, etc. I understand the page limitation constraint, but including such baselines in future work would make the experimental results more convincing and provide a more comprehensive evaluation.

**Detailed Comments:**

No further comment.

**Justification Of Final Rating:**

The authors did their best to address my comments within the given 1-week timeframe. If all of my suggested points are incorporated in a future study or an extended journal version, I believe this could become a more promising piece of work. However, in its current form, I am unable to further improve the rating of the paper.

**Justification Of The Preliminary Rating:**

The proposed methodology is technically sound. While individual components, such as Monte Carlo dropout for uncertainty estimation and weighted least squares optimization, are based on established techniques, their integration into the overall pipeline is well reasoned. Applying them wisely in the keypoint detection phase within the 2D/3D registration, combined with sufficient experimental evaluation, makes the approach seemingly incremental yet sufficiently novel for MIDL.

**Questions To Address In The Rebuttal:**

See the Weaknesses section above.

---

> ### Author Response · Authors · 2026-01-24
> **Response for Reviewer Bdjs**
>
> We thank you for your time and constructive feedback.
>
> ---
> **Reviewer** `Bdjs` suggested connections to learned keypoint discovery.
> **Our Reponse**:
> We thank the reviewer for bringing the work of Ruhaak et al. to our attention. We were previously unaware of this specific study. We strongly agree with the perspective that pre-defined anatomical landmarks, which are typically established based on physical dissection or semantic importance, are often not the most coherent or distinct features. Consequently, we view the transition toward automatically discovered learned landmarks as a critical future direction for this framework.
>
> ---
> **Reviewer** `Bdjs` raised concerns about the fixed landmark removal threshold and its impact on generalization.
> **Our Reponse**:
> The threshold $K$ was determined empirically using a separate validation set to maximize the trade-off between outlier rejection and pose estimation accuracy. We agree that automatic hyper-parameter estimation would be useful, but we also believe that these methods will largely rely on the same validation-holdout schema to set those parameters. Thus, derivation of an analytic criterion seems very useful but difficult and potentially fragile (similar to selecting the number of clusters in a clustering method, or automatic relevance determination [ARD]). If we find that an automatic threshold determination method is feasible and performant, we will validate it in future work.
>
> ---
> **Reviewer** `Bdjs` questioned the validity of uncertainty estimates when evaluated only on synthetic data.
> **Our Reponse**:
> We fully agree that validation on real clinical or cadaveric data is the ultimate standard for ensuring translational utility. However, as a large-scale dataset with paired intra-operative fluoroscopy and accurate ground-truth 3D pose was not available for this work, we utilized high-fidelity synthetic data as a necessary first step to establish the methodological validity of our uncertainty framework.
>
> ---
> **Reviewer** `Bdjs` suggested including additional classical baselines such as RANSAC-based PnP.
> **Our Reponse**:
> RANSAC is indeed a robust standard for hard outlier rejection in image-based feature matching, and there are proposed RANSAC based fluoroscopy methods in the literature. We will attempt to implement these for future experiments, though the limited number of views (two per instance) and the fluoroscope/x-ray modality means that it’s unlikely that out-of-the-box RANSAC/Iterative Closest Point will work well.
> Our method also adopts a slightly different approach, where each landmark has known correspondence, and we want to quantify the spatial uncertainty of its estimate.  However, we recognize both seek to solve the same problem and will further consider their individual advantages and differences in future work.

---

### Official Review · Reviewer_K2ZM · 2026-01-16

**Confidence:** 3
**Preliminary Rating:** 4
**Final Rating:** 4

**Summary:**

This paper proposes an uncertainty-aware registration framework to model predicted landmark uncertainty. The framework is tested on synthetic fluoroscopy from a publicly available pelvic CT dataset. The results support that this method helps identify unreliable landmarks and stabilize 2D/3D pose estimation.

**Strengths:**

1. The authors propose a differentiable uncertainty-to-weight formulation that allows continuous weighting and show selection strategies that improve robustness without re-training.
2. The structure and language is excellent, and the equations and figures are well designed and annotated.
3. The experiments are well designed and extensive, exemplified by including an error retention analysis for validation.
4. Prior work is addressed explicitly in section 2, related work, and it cites relevant work across the topics of registration, uncertainty estimation, and statistical methods.
5. Data sources and prior results are cited appropriately, and the included repository link with code and instructions indicate a good level of reproducibility.

**Weaknesses:**

1. Maybe I overlooked it somehow, but it would be nice to see more details about the training scheme for the UNet. What are the number of samples used in training, validation, testing?
2. Could you please also report the spread of errors along with the mean/median results? I see several outliers in the figures 2, 3 and 4, but the tables indicate only the mean and median. It could help to understand if the standard deviations of comparable methods are also higher than the proposed method.
3. No action is expected based on this comment – but I imagine demonstrating the generality of this method on more than one data set could make the case for it’s utility stronger.

**Detailed Comments:**

Here are some requests for clarification and minor improvements:

1. In the Introduction, you state: “While this has the potential to have high accuracy and generality across anatomy, each forward pass is generally computationally expensive and thus often slow for bed-side applications. Landmark methods instead rely on anatomic knowledge of the target volume, and match pre-defined features or landmarks between the 2D and 3D sets. While this is much more computationally tractable, avoiding the reprojection steps of intensity matching, it is prone to higher errors due to the sensitivity of the point matching operation, and due to the intrinsic variability of anatomic landmarks.”
This gives me the impression that the baseline Landmark based methods must be less accurate than the baseline Intensity matching method. However, in Table 1, I see the intensity matching methods are worse across the board. Could you clarify this? Is this setting/dataset not really amenable for intensity matching methods?

2. In section 3.1, it is mentioned that “To improve generalization, we utilize a dilation-erosion label augmentation scheme”. Would you think it is useful to explore, and if so, run ablation studies showing what the effect of this choice is? The same for this: “β as a hyper-parameter controlling weight “fall-off”, which will correspond to outlier suppression strength in the resulting optimization”. How sensitive is translation and rotation errors to the choice of β?

3. In the experiment for discrete selection, I understand that there are originally 14 landmark points, and progressively removing up to K=7 points shows improvements in errors. Since the errors continue to reduce till the end of the plot, I wonder if you could show at what point the errors start rising again?

**Justification Of Final Rating:**

The authors addressed all the comments that could be addressed in the time frame provided to them. Weighing the strengths and limitations of the approach as described, I am more confident now of the preliminary rating, which based on all the other manuscripts I have read in this format, appears to be a fair assessment.

**Justification Of The Preliminary Rating:**

This paper makes a strong case for uncertainty-weighted selection of landmark keypoints for more robust pose estimation, through sound theoretical choices, and extensive experimental results, albeit on a single data set. With minor clarifications and improvements, this would certainly be of interest to the larger MIDL community.

**Questions To Address In The Rebuttal:**

Please see the comments in the weaknesses and detailed comments section for updates that could help clarify and improve the paper further.

---

> ### Author Response · Authors · 2026-01-24
> **Response for Reviewer K2ZM**
>
> We thank you for your time and constructive feedback.
>
> ---
> **Reviewer** `K2ZM` requested additional details on the model architecture and dataset construction.
> **Our Reponse**:
> We apologize for omitting the details regarding the model structure and dataset construction.  Regarding the model, we employed the dilation-erosion pipeline proposed in Suh et al. [1], which has demonstrated strong generalization capabilities in diverse orthopedic imaging contexts. This method initializes training by dilating landmark annotations into larger regions to help the network learn coarse localization, and then progressively erodes these labels back to their original size over a scheduled number of epochs. This creates a dynamic curriculum that refines spatial accuracy as the model converges. Concerning the dataset, we generated 600 Digitally Reconstructed Radiographs (DRRs) for each subject. These were split into training, validation, and testing sets using a 75:15:10 ratio, corresponding to 450, 90, and 60 images per subject, respectively. We have augmented the paper to include these relevant details.
>
> [1] Suh, Y., Chan, P., Martin, J. R., & Moyer, D. (2023). Label Augmentation Method for Medical Landmark Detection in Hip Radiograph Images. arXiv preprint arXiv:2309.16066.
>
> ---
> **Reviewer** `K2ZM` suggested reporting the spread of errors in addition to mean and median values.
> **Our Reponse**:
>
> |     Experiment     | Rotation Error |  P=50 |  P=60 |  P=70 |  P=80 |  P=90 |
> |:------------------:|:------------------------------------:|:-----:|:-----:|:-----:|:-----:|:-----:|
> |     No Weights     |            26.14 $\pm$ 30.08           | 12.96 | 17.45 | 29.92 | 41.46 | 75.48 |
> |         DS         |            14.22 $\pm$ 29.18           |  2.31 |  3.39 |  5.42 | 13.68 | 51.83 |
> | Finetune + NG + DS |            16.52 $\pm$ 30.38           |  2.87 |  4.51 |  9.24 | 21.09 | 57.17 |
> |    Finetune + DS   |            16.33 $\pm$ 30.78           |  2.84 |  4.37 |  8.08 | 20.00 | 59.39 |
> |         CW         |            13.94 $\pm$ 28.67           |  2.27 |  3.43 |  5.36 | 12.72 | 46.68 |
> | Finetune + NG + CW |            15.65 $\pm$ 28.57           |  2.73 |  4.45 |  9.23 | 20.65 | 54.13 |
> |    Finetune + CW   |            15.84 $\pm$ 30.80           |  2.73 |  4.78 |  7.41 | 16.97 | 49.39 |
>
> |     Experiment     | Translation Error |  P=50 |  P=60 |  P=70 |  P=80 |  P=90  |
> |:------------------:|:-----------------:|:-----:|:-----:|:-----:|:-----:|:------:|
> |     No Weights     | 51.18 $\pm$ 56.58 | 32.70 | 45.09 | 64.30 | 80.86 | 111.67 |
> |         DS         | 24.79 $\pm$ 50.58 | 6.38  | 9.09  | 13.59 | 33.46 | 70.17  |
> | Finetune + NG + DS | 21.98 $\pm$ 33.38 | 7.46  | 11.21 | 21.61 | 33.64 | 60.30  |
> |    Finetune + DS   | 21.92 $\pm$ 36.44 | 7.21  | 10.53 | 17.19 | 32.08 | 55.74  |
> |         CW         | 24.18 $\pm$ 62.23 | 6.35  | 8.98  | 12.96 | 27.99 | 58.35  |
> | Finetune + NG + CW | 22.14 $\pm$ 34.78 | 7.44  | 10.78 | 20.02 | 32.67 | 58.41  |
> |    Finetune + CW   | 20.63 $\pm$ 32.54 | 6.97  | 10.24 | 18.84 | 30.99 | 55.14  |
>
> Thank you for this suggestion. We have added the standard deviation alongside error values at the 50th, 60th, 70th, 80th, and 90th percentiles to provide a more comprehensive view of the error distribution. Note that since the translation errors are reported in millimeters, the observed spread represents relatively minor physical deviations in the context of pelvic surgery even in the higher percentiles.
>
>
> ---
> **Reviewer** `K2ZM` questioned the apparent discrepancy between landmark-based and intensity-based baselines.
> **Our Reponse**:
> We apologize for the confusion. The lower performance of intensity-based methods in Table 1 stems from their high sensitivity to initialization (so called “capture region” [2]). Intensity-based registration often fails when the initial misalignment is large due to non-convex optimization landscapes. Our test set includes large initial perturbations (rotations up to $\pm$ 45 degrees and translations up to $\pm$ 50 mm), which could leave the capture region leading to divergence.
>
> [2] Unberath, M., Gao, C., Hu, Y., Judish, M., Taylor, R. H., Armand, M., & Grupp, R. (2021). The impact of machine learning on 2d/3d registration for image-guided interventions: A systematic review and perspective. Frontiers in Robotics and AI, 8, 716007.

---

> ### Author Response · Authors · 2026-01-24
> **Response for Reviewer K2ZM**
>
> **Reviewer** `K2ZM` asked about ablation studies and sensitivity to the dilation–erosion scheme and β hyperparameter.
> **Our Reponse**:
> Regarding the dilation-erosion scheme, we utilized the method established in Suh et al. [3] as a fixed backbone because it has already been proven to robustly generalize in similar orthopedic imaging tasks, and thus we focused our ablation studies on our pipeline, which sits on top of this dilation-erosion framework.
> Concerning the $\beta$ hyperparameter, while we agree that a sensitivity analysis would be valuable to characterize the precise relationship between outlier suppression and error, conducting a comprehensive sweep is computationally expensive and unfortunately infeasible within the rebuttal timeframe. However, we view this as an important experimental result and will add this to a potential camera-ready submission.
>
> [3] Suh, Y., Chan, P., Martin, J. R., & Moyer, D. (2023). Label Augmentation Method for Medical Landmark Detection in Hip Radiograph Images. arXiv preprint arXiv:2309.16066.
>
> ---
> **Reviewer** `K2ZM` inquired whether performance degrades when removing more than seven landmarks.
> **Our Reponse**:
> The continued error reduction observed up to $K=7$ indicates that our uncertainty metric successfully identifies and filters the least reliable landmarks. However, exploring values beyond $K=7$ encounters a hard constraint due to landmark availability. Since not all 14 landmarks are visible in every fluoroscopic image because some landmarks fall outside the imaging field of view and are therefore cropped by the image bounding box, aggressive filtering significantly reduces the number of usable points. In our dataset, removing more than 7 landmarks results in scenarios where fewer than 3 points remain (occurring in 37.4% of cases after $K=7$), rendering the Perspective-n-Point (PnP) solver undetermined. Consequently, the effective limit for $K$ is dictated by the minimum point requirement for the PnP algorithm rather than a degradation in the uncertainty metric's capability. We have expanded our discussion to clarify this constraint.

---

> ### Comment · Reviewer_K2ZM · 2026-01-26
> **Thank you for the clarifications**
>
> Dear Authors,
> You have addressed all the addressable (within this time frame) points that were raised. I have nothing more to add at this moment. I wish you the best in the rest of the review process, and look forward to read the revised manuscript.

---

### Meta-Review · Area_Chair_dMuw · 2026-02-06

**Recommendation:** Accept (Poster)
**Confidence:** 5

**Metareview:**

Overall, the reviewers’ recommendations lean toward acceptance after rebuttal. While there are some concerns that the proposed technical components may not be highly novel, as well as moderate concerns regarding the experiments on the 3D synthetic datasets, the motivation and the practical significance of the work are clear. The proposed uncertainty-aware, landmark-based 2D/3D pelvic pose estimation framework holds promising value to improve image-guided intervention workflows. Given the strong practical relevance, well designed experiments, and potential clinical impact, the meta-reviewer is pleased to recommend acceptance for MIDL.

---

### Decision · Program_Chairs · 2026-02-13

Accept (Poster)